# Monte Carlo Study of Electronic Transport in Monolayer InSe

**DOI:** 10.3390/ma12244210

**Published:** 2019-12-14

**Authors:** Sanjay Gopalan, Gautam Gaddemane, Maarten L. Van de Put, Massimo V. Fischetti

**Affiliations:** Department of Materials Science and Engineering, The University of Texas at Dallas, Richardson, TX 75080, USA; sxg174530@utdallas.edu (S.G.); gautam.gaddemane@utdallas.edu (G.G.);

**Keywords:** InSe, phonon scattering, intrinsic transport, DFT, Density Functional Perturbation Theory, mobility, Monte Carlo

## Abstract

The absence of a band gap in graphene makes it of minor interest for field-effect transistors. Layered metal chalcogenides have shown great potential in device applications thanks to their wide bandgap and high carrier mobility. Interestingly, in the ever-growing library of two-dimensional (2D) materials, monolayer InSe appears as one of the new promising candidates, although still in the initial stage of theoretical studies. Here, we present a theoretical study of this material using density functional theory (DFT) to determine the electronic band structure as well as the phonon spectrum and electron-phonon matrix elements. The electron-phonon scattering rates are obtained using Fermi’s Golden Rule and are used in a full-band Monte Carlo computer program to solve the Boltzmann transport equation (BTE) to evaluate the intrinsic low-field mobility and velocity-field characteristic. The electron-phonon matrix elements, accounting for both long- and short-range interactions, are considered to study the contributions of different scattering mechanisms. Since monolayer InSe is a polar piezoelectric material, scattering with optical phonons is dominated by the long-range interaction with longitudinal optical (LO) phonons while scattering with acoustic phonons is dominated by piezoelectric scattering with the longitudinal (LA) branch at room temperature (T = 300 K) due to a lack of a center of inversion symmetry in monolayer InSe. The low-field electron mobility, calculated considering all electron-phonon interactions, is found to be 110 cm^2^V^−1^s^−1^, whereas values of 188 cm^2^V^−1^s^−1^ and 365 cm^2^V^−1^s^−1^ are obtained considering the long-range and short-range interactions separately. Therefore, the calculated electron mobility of monolayer InSe seems to be competitive with other previously studied 2D materials and the piezoelectric properties of monolayer InSe make it a suitable material for a wide range of applications in next generation nanoelectronics.

## 1. Introduction

The great success of graphene [1,2] has been followed by an equally impressive surge of the study of other two-dimensional (2D) materials. Recently, 2D materials, including graphene [1,2], phosphorene [3,4,5,6,7], silicene [8,9,10], silicane [9,11,12,13], germanene [8,9,10,14], and transition metal dichalcogenides [15,16,17,18,19,20], have been widely studied for their unique electrical and optical properties. The presence of a band gap in 2D transition-metal chalcogenides [21] has made this class of materials appealing for applications in field effect transistors (FETs).

The intrinsic thermodynamic instability of 2D materials, which originates from Mermin–Wagner theorem [22,23], has been challenged on theoretical grounds. This instability stems from the parabolic dispersion of the acoustic flexural (“out-of-plane” or ZA) modes which causes their thermal population to diverge at a long wavelength at any finite temperature. In 2D materials like silicene and germanene, that lack horizontal mirror ($\sigma_{h}$) symmetry, Fischetti et al. [24] and Gaddemanne et al. [8] have shown that the coupling of electrons to the ZA phonons is extremely strong, an effect that results in extremely low mobilities [24]. However, 2D $\sigma_{h}$-symmetric crystals, such as monolayer InSe, are immune to this problem since the scattering potential associated with the flexural displacement is odd with respect to $\sigma_{h}$. Thus, the electron-phonon matrix element vanishes and intraband electronic transitions assisted by these flexural modes are forbidden to first order [24].

InSe exists in three polytypes, namely, β, γ, and ε-InSe. γ-InSe exhibits an ABCABC stacking and lacks intrinsic stability; this hinders its application in electronics [25,26]. The ε-InSe has ACAC layer arrangement and exhibits indirect band structure with high photoresponsivity. β-InSe has the same composition as γ-InSe but differs in its stacking configuration and it is notable among the three polytypes since it has been exfoliated into individual layers with honeycomb structure [27] and is the most stable phase of InSe thanks to its ABAB stacking configuration [28]. However, in monolayer form, all three polytypes have the same crystal structure [29], which we simply call monolayer InSe. Recent developments of liquid-phase exfoliation of InSe flakes [30] and several emerging growth techniques [31] make InSe a material of interest for future complementary metal-oxide-semiconductor technology.

Recently, contradictory results have been presented on the calculated mobility in few-layer InSe. Li et al. [32] have calculated the intrinsic electron mobility in monolayer, bilayer, and bulk β-InSe, reporting a mobility of 120 cm^2^V^−1^s^−1^ in monolayers. The electron-phonon matrix elements were calculated using density functional theory (DFT) and the Boltzmann transport equation (BTE) and solved in the self-energy relaxation time approximation (SERTA) obtaining an electron mobility similar to the results we shall present below. Wang et al. [29] performed a study of the thermoelectric transport properties of monolayer InSe calculating the mobility within the constant relaxation-time approximation. They predicted the electron mobility in monolayer InSe to be 1300 cm^2^V^−1^s^−1^. However, the use of constant deformation potential fails to account for the anisotropy of the matrix elements, leading to an overestimation of the mobility. Gaddemane et al. [3] have shown the importance of selecting the correct physical models and numerical approximations, as the lack of an accurate treatment has led to overestimation of mobility, as we suspect is the case in [29].

In this paper, we present a theoretical study of the electronic-transport properties of monolayer InSe. The electronic structure, the phonon dispersion, and the carrier-phonon matrix element of monolayer InSe are determined using DFT in combination with density functional perturbation theory (DFPT). The electron-phonon scattering rates are calculated using Fermi’s Golden Rule. ZA-phonon-scattering is ignored since its $\sigma_{h}$-symmetry prevents electrons from coupling at first order to these flexural modes. The BTE is solved using the full-band Monte Carlo method to calculate the low- and high-field characteristics.

## 2. Structure and Computational Methodology

### 2.1. Structure

InSe is a layered chalcogenide semiconductor. Each layer exhibits a honeycomb (or hexagonal) lattice consisting of four atomic planes of covalently bonded Se-In-In-Se, as shown in Figure 1. The 2-H layered structure of monolayer InSe is symmetric under reflections on the horizontal plane of the crystal ($\sigma_{h}$).

The structural and electronic parameters for monolayer InSe are listed in Table 1 and are in good agreement with previous work [29,33,34,35].

The lattice constant and the bond lengths are obtained by relaxing the initial assumed structure [36] using DFT by minimizing the total energy of the structure. The calculation of band gap is explained briefly in Section 2.2.1 and presented in Section 3.1.

### 2.2. Theoretical Model

In this section, we briefly present the computational details and provide the key components of our theoretical model. Additional information is available in [7].

#### 2.2.1. Band Structure and Phonon Spectrum

The band structure of monolayer InSe is obtained from density functional theory, as implemented in the Quantum ESPRESSO (QE) [37,38] software package, with both the Perdew–Burke–Enzerhoff generalized-gradient approximation (GGA-PBE) [39] and local density approximation (LDA) [40] for the exchange-correlation functional, and the ONCV pseudopotentials [41]. The total energy of the structure is minimized by relaxing the lattice vectors and atomic positions until the final force on each atom is less than 10^−4^ eV/nm. A vertical spacing of approximately 2.5 nm is used between adjacent layers, a value large enough to minimize artifacts arising from the interactions between adjacent supercells. The computational parameters used in the calculations are shown in Table 2. The phonon spectrum for the system is obtained using QE which implements density functional perturbation theory (DFPT) [42]. We initially calculate the phonon energies using a coarse ***k*** (12 × 12 × 1) and ***q*** (6 × 6 × 1) mesh with a self-consistent threshold of 10^−16^ Ry to avoid nonphysical imaginary frequencies for the low-energy acoustic phonons and interpolated on a fine ***k*** (30 × 30 × 1) and ***q*** (30 × 30 × 1) mesh using maximally localized Wannier functions, as implemented by the Electron-phonon Wannier (EPW) package [43].

#### 2.2.2. Electron-Phonon Interaction and Scattering Rates

The electron-phonon scattering rates are calculated to first order using Fermi’s Golden Rule. The matrix element for an electron in an initial state with wave vector ***k*** in band *m* to a final state with wave vector ***k* + *q*** in band *n* is given by
(1)$$g_{mn}^{\nu }\left(k,q\right)=\sqrt{\frac{1}{2\omega_{\nu ,q}}}\langle \psi_{n,k+q}|\partial V_{q,\nu }^{\mathrm{SCF}}|\psi_{m,k}\rangle$$
where $\partial V_{q,\nu }^{\mathrm{SCF}}$ is the change of the self-consistent Kohn–Sham potential due to the presence of a phonon of wave vector ***q***, branch (longitudinal or transverse, acoustic or optical) index ν, and frequency $\omega_{\nu ,q}$. $\psi_{n,k}$ is the electronic wavefunction for band *n* and wavevector ***k***.

The matrix elements $g_{mn}^{\nu }\left(k,q\right)$ can be calculated from DFPT using QE. However, the evaluation of $g_{mn}^{\nu }\left(k,q\right)$ on an extremely dense mesh in the first Brillouin zone (BZ) is computationally prohibitive. This problem has been overcome by an interpolation strategy based on maximally localized Wannier functions (MLWF). We use the EPW [43] code of the QE suite to obtain the electron-phonon matrix elements on a fine mesh in the first Brillouin zone, similar to calculating the phonon energies, as discussed in Section 2.2.1.

Since InSe is a polar material, we expect the coupling of electrons to the long-wavelength LO phonons to be strong [44]. In order to better understand its importance, we separate the short-range ($g^{S}$) and long-range ($g^{L}$) contributions to the matrix elements as
(2)$$g_{mn}^{\nu }{\left(k,q\right)}^{2}={\left[g_{mn}^{\nu ,S}\left(k,q\right)+\;g_{mn}^{\nu ,L}\left(k,q\right)\right]}^{2}$$

For a given phonon with wavevector ***q*** belonging to branch ν, the long-range contribution, $g^{L}$, to the matrix element is given by [45]
(3)$$g_{mn}^{\nu ,L}\left(k,q\right)=i\frac{e^{2}}{\Omega \varepsilon_{0}}\underset{a}{\sum }\sqrt{\frac{\hbar }{2NM_{a}\omega_{\nu ,q}}}\underset{G\neq -q}{\sum }\frac{\left(q+G\right)\cdot Z_{k}^{*}\cdot e_{k\nu }\left(q\right)}{\left(q+G\right)\cdot \varepsilon^{\infty }\cdot \left(q+G\right)}\;\langle \psi_{n,k+q}|e^{i\left(q+G\right)\cdot r}|\psi_{m,k}\rangle$$
where Ω is the volume of the unit cell; *M_a_* is the mass of atom ‘a’; *N* the number of unit cells; $Z_{k}^{*}$ is the Born effective charge tensor; $e_{k\nu }\left(q\right)$ represents a vibrational polarization vector normalized within the unit cell; ***G*** indicates the reciprocal lattice vectors; and e, $\varepsilon_{0}$, $\varepsilon^{\infty }$, and ℏ are the electron charge, static permittivity, high-frequency permittivity tensor, and reduced Plank constant, respectively. Under the assumptions of [46], $g^{L}$ reduces to the usual Fröhlich expression. 

The short-range contribution is defined as
(4)$$g_{mn}^{\nu ,\mathrm{Short}}{\left(k,q\right)}^{2}=g_{mn}^{\nu }{\left(k,q\right)}^{2}-g_{mn}^{\nu ,L}{\left(k,q\right)}^{2}$$
where $g_{mn}^{\nu ,\mathrm{Short}}{\left(k,q\right)}^{2}=g_{mn}^{\nu ,S}{\left(k,q\right)}^{2}+2g_{mn}^{\nu ,S}\left(k,q\right)g_{mn}^{\nu ,L}\left(k,q\right)$ and the second term on the right-hand side is the linear long-range contribution. 

The rate at which an electron with wavevector ***k*** in band *m* scatters by either emitting or absorbing a phonon with wavevector ***q*** of branch ν is
(5)$$\frac{1}{\tau_{m}^{\nu }\left(k\right)}=\frac{2\pi }{\hbar }\underset{q,n}{\sum }{\left|g_{mn}^{\nu }\left(k,q\right)\right|}^{2}\left[N_{\nu ,q}+\frac{1}{2}\pm \frac{1}{2}\right]\delta \left[E_{m,k}-E_{n,k+q}\pm \hbar \omega_{\nu ,q}\right]$$
where $N_{\nu ,q}$ is the Bose–Einstein phonon occupation at temperature T.

The presence of the energy-conserving Dirac-delta function in Equation (5) is handled numerically using the Gilat–Raubenheimer algorithm [47].

#### 2.2.3. Monte Carlo Simulations

The quantities that are required as an input to the Monte Carlo simulations are the electronic band structure, the phonon frequencies, and the electron-phonon matrix elements. The band structure is calculated and tabulated on a very fine mesh of 201 × 201 × 1 ***k***-points, covering the upper-right rectangular section of the Brillouin zone. This rectangular section covers the irreducible triangular wedge and an additional region that completes the rectangle. The mesh is fine enough to ensure a good resolution of the carrier velocity for mobility calculations. The phonon frequencies and the electron-phonon matrix elements are tabulated on a fine ***k*** (30 × 30 × 1) and ***q*** (30 × 30 × 1) mesh. They are further interpolated on the band-structure mesh using bilinear interpolation.

Having obtained the electron-phonon scattering rates, we solve the Boltzmann transport equation stochastically using a full-band Monte Carlo method by simulating an ensemble of 500 particles with time steps of 0.1 fs until steady-state is obtained, this method has been previously described and used in [21] to study electronic transport in silicene and germanene. The Monte Carlo calculations of the low-field mobility are performed assuming a zero electric field and estimating the electron mobility $\mu_{\Theta }$ along the direction θ from the diffusion constant $D_{\theta }$, using the Einstein relation. This method provides results that are affected by a reduced statistical noise when compared with estimates of the mobility from the linear (low-field) region of the velocity-field characteristics [48]. A uniform electric field is assumed when calculating the high-field behavior.

## 3. Results

### 3.1. Electronic Band Structure and Phonon Spectrum

In Figure 2a, we show the calculated band structure of monolayer InSe along the high-symmetry directions in the first Brillouin zone. Monolayer InSe is an indirect band gap material with a band gap of 1.45 eV, with the conduction-band minimum at Γ, and valence-band maxima along the Γ-K direction. We have found that the effect of the spin-orbit coupling (SOC) on the band structure is negligible and it does not change the electron effective mass of the material. Therefore, we have not included the effect of the SOC in our final results presented below. The rotational symmetry of the equi-energy contours close to the conduction-band minimum and the parabolic nature of band structure at Γ implies an isotropic electron effective mass of about 0.18 m_e_. Figure 2b illustrates the phonon dispersion of the 12 phonon branches that result from the presence of 4 atoms in the unit cell. The three lowest-energy branches at Γ are the ZA phonons, the longitudinal acoustic (LA), and the transverse acoustic (TA) phonons, respectively. The optical branches corresponding to longitudinal optical phonons have a maximum frequency away from the Γ point in the Brillouin zone. This is a well-known phenomenon, known as ‘overbending’ [49] or ‘Kohn anomaly’ [50], that has been previously discussed in the case of monolayer h-BN [49].

### 3.2. Scattering Rates

In Figure 3, we show the room-temperature scattering rates as a function of initial electron energy (average over equi-energy surfaces) calculated including both the long-range and the short-range interactions, as well as their individual contributions. As expected, Figure 3a shows that the total scattering rate for scattering with optical phonons is about twice as large as the total rate for scattering with all acoustic branches. In particular, the contribution of the LO phonons is about 2 orders of magnitude higher than the contribution by all other optical modes. This is due to the polar nature of monolayer InSe that causes a very strong long-range Fröhlich interaction, as discussed in Refs. [32,44].

However, InSe exhibits a feature that seems rather unusual. Figure 3b shows the presence of a large scattering rate with acoustic phonons even when isolating only the long-range contribution. This strong interaction is due to the strong piezoelectric properties of monolayer InSe in combination with a lack of a center of inversion symmetry [51]. Indeed, for a typical III-V compound semiconductor at T = 300 K, the effect of piezoelectric scattering on the mobility is about 3 to 4 orders of magnitude lower than the deformation potential interaction. Therefore, piezoelectric scattering is usually not as important as acoustic phonon scattering due to deformation potential or ionized impurity scattering. However, piezoelectric scattering can become important for group III monochalcogenides since the crystal structure lacks inversion symmetry and hence, the piezoelectric stress tensor does not vanish [52]. Therefore, piezoelectric scattering in monolayer InSe is important even at high temperatures, in contrast to III-V compound semiconductors. The contribution of the short-range interaction to the scattering rate is about one order of magnitude smaller than contribution of the long-range interactions, as seen in Figure 3c, and it does not affect the final transport properties of the material. 

In Figure 3, one can see the onset of LO-phonon-emission processes at an electron kinetic energy of about 25 meV, the energy of the LO-phonon itself. In 2D materials like TMDs [18] and phosphorene [3], there is an additional steplike increase in the electron scattering rate, denoting the onset of intervalley scattering. Such step is not seen in monolayer InSe, since the Γ-M intervalley scattering matrix elements are smaller than the strong intravalley matrix elements in the Γ-valley.

In Figure 4, we show the calculated electron-phonon matrix elements at low-energy as a function of the angle between the initial and final wavevector around the Γ symmetry point for various phonon branches. The isotropic nature of the electron-phonon matrix elements when plotted as a function of scattering angle can be observed in Figure 4. We also found that the third acoustic branch (Figure 4) and the 11th branch (an optical phonon, OP8) dominate transport. The dominant scattering mechanism is intravalley scattering in the Γ-valley, since the next local minimum at the M point lies at an energy approximately 0.7 eV higher. Therefore, this Γ-M intervalley process affects high-field electron transport but not the low-field mobility.

### 3.3. Mobility Calculations

Recently, in [53], it has been shown that, at least in some cases, different choices of exchange-correlation functionals and even pseudopotentials can result in vastly different results for the carrier mobility calculated using DFT. Therefore, we have repeated our calculations using the same ONCV pseudopotentials using the local-density approximation (LDA) exchange functional in lieu of GGA-PBE in order to assess the sensitivity of our results to the different “flavors” of DFT used. We have not used nonlocal exchange correlation functionals such as PBE0 or HSEO6 for the band structure calculations. Although, hybrid functionals give more accurate band structures compared to what we obtain using LDA or GGA-PBE. However, at present, the calculation of the electron-phonon matrix elements is not supported by available software packages when using these hybrid functionals. In transport calculations, especially when we do not consider tunneling or bipolar devices, electron-phonon matrix elements and band curvature are much more important than the band-gap issue. Regarding the ‘bigger issue’, (namely, *is the use of any functional, hybrid or not, that yields a ‘better’ band structure, going to yield also more accurate electron-phonon matrix elements?*) we think that the answer is a ‘probably not’, given the degree of empiricism embedded in all models of the exchange and correlation functionals and the ‘pseudization’ of the wavefunctions and given our past experience [53].

In Table 3, we summarize the low-field mobility as obtained from the diffusion constant extracted from Monte Carlo simulations, as discussed before. We show the low-field electron mobility including all processes, as well as the mobility limited by the long- and short-range interactions separately.

The increase in the electron mobility calculated using the LDA exchange-correlation functional, when compared to GGA-PBE, is attributed to the low scattering rates (Figure 3). Note that this difference arises only from the choice of the exchange-correlation used to perform the DFT calculations. Nevertheless, the difference in the electron mobility between the two cases discussed above is much smaller than what was found in the case of WS_2_ [53]. The weaker dependence on the choice of the exchange-correlation functional of the calculated mobility in InSe, compared to the case of WS_2_, is due to the smaller role played by intervalley scattering in InSe than in WS_2_. Therefore, differences in the values of the intervalley energetic separation that result from different “flavors” of DFT affects to a much smaller extent the calculation of the electron mobility.

### 3.4. Velocity-Field Calculations

Finally, we discuss the velocity-field characteristic obtained using the GGA-PBE exchange-correlation functional (GGA-PBE case) and compare the results with LDA functional (LDA case). In Figure 5a,c, we show the drift velocity of electrons under the effect of a homogeneous applied electric field along the zigzag direction as well as their average kinetic energy for the GGA-PBE case. We show with the dashed red line the drift velocity obtained from the mobility calculated at zero field from the diffusion constant. The results of the two methods are in good agreement, as seen in Figure 5a. However, at high fields, above about 10^4^ V/cm, the onset of electron heating induces a saturation of the electron velocity. Due to the significant increase of the average energy, the electrons in the Γ-valley gain enough energy to scatter to the M-valley, an effect that reduces the mobility at high fields+ and even results in negative differential mobility, as seen in Figure 5a,c. We show, in Figure 5b,e, the velocity-field characteristics and average energy of electrons, accounting only for the long-range interactions. Figure 5c,f show the same quantities calculated when considering only the short-range interactions.

The steady-state electron distribution in ***k***-space for the GGA-PBE case is shown in Figure 6 for various values of the strength of the applied electric field. At low fields, in Figure 6a, all the electrons populate only the region near the conduction-band minimum in the Γ-valley. However, at high fields, the electrons gain enough energy to scatter to the M-points (Figure 6c). This results in the saturation of the drift velocity at a field strength larger than about 10^4^ V/cm, resulting also in negative differential mobility for fields exceeding 4 × 10^4^ V/cm, due to low electron velocity in the M-valleys. The low electron velocity in the M-valley is attributed to its high effective mass (0.45 m_e_ along M-K direction and 1.5 m_e_ along M-Γ direction) compared to the Γ-valley (isotropic effective mass of 0.18 m_e_).

In the LDA case, we observe a similar saturation of the drift velocity (Figure 5). However, compared to the GGA-PBE case, we see more electrons populating the M-valley for the LDA case (Figure 6e,h) at an applied field of 4 × 10^4^ V/cm. This is due to the low energy difference between the minima at Γ-point and the M-point using LDA (0.6 eV) compared to GGA-PBE (0.74 eV). Therefore, we observe a very sharp negative differential mobility for fields exceeding 4 × 10^4^ V/cm in LDA (Figure 5).

## 4. Discussion

In order to assess the potential of monolayer InSe as a channel material in high-performance field-effect devices, we compare the intrinsic mobility of monolayer InSe we calculate with other previously studied 2D channel materials. Theoretical studies have predicted a mobility ranging from 70 to 410 cm^2^V^−1^s^−1^ for monolayer MoS_2_ [15,16,17,18,19,20]. However, the ‘best-case scenario’ prediction of an electron mobility of 410 cm^2^V^−1^s^−1^ [17] results from the assumption of an electron-phonon matrix element independent of initial wave vector and using constant deformation potential, thus overestimating the mobility. An approach similar to ours has been followed in Ref. [18] that reports an electron mobility of about 130 cm^2^V^−1^s^−1^, a value quite similar to our result for monolayer InSe. Values of 25, 30, and 320 cm^2^V^−1^s^−1^ have been reported for monolayer MoSe_2_, WS_2_, and WSe_2_, respectively [18]. Phosphorene, another well-studied 2D material, has been predicted to exhibit an electron mobility of 10 and 21 cm^2^V^−1^s^−1^ [3] along the zigzag and armchair directions, respectively. When comparing the theoretically predicted electron mobility of other 2D materials with monolayer InSe, we can conclude that monolayer InSe is a promising channel material for future FETs.

On the other hand, the electron mobility of intrinsic bulk Si is known to be about 1450 cm^2^V^−1^s^−1^ [54]. However, in Si thin layers, the electron mobility decreases drastically with decreasing thickness of the layer. The dielectric constant of monolayer InSe is about 3.88 [55], and the thickness of the monolayer is 0.55 nm. Therefore, the physical thickness of a Si film with a dielectric thickness similar to monolayer InSe is about 1.7 nm (multiplying by a factor of 11.9/3.88). For such a thin Si layer, the phonon-limited electron mobility is about 500 cm^2^V^−1^s^−1^ under a transverse (gate) field of 10^5^ V/cm. An even more severe reduction of the electron mobility in Si thin slabs is caused by scattering with surface roughness. This can depress the electron mobility to 350 cm^2^V^−1^s^−1^ or even less than 200 cm^2^V^−1^s^−1^ [56], depending on the nature of the roughness. Therefore, monolayer InSe exhibits promising transport properties, even when compared to equivalently thin Si slabs.

Experimental studies have demonstrated that monolayer InSe has a far superior photoresponsivity and faster response time than MoS_2_ monolayers [57]. Its good piezoelectric properties, combined with the excellent photosensitivity of monolayer InSe, enable a wide range of application such as stress-sensor, photodetectors, and possibly nanoscale transducers [51].

## 5. Conclusions

We have shown DFT predicts a rather large band gap of about 1.45 eV for monolayer InSe, making it suitable for complementary-logic applications. However, whereas the conduction band exhibits a conventional-looking minimum at the Γ-point, the valence band maxima are away from Γ-point, so that the structure of the upper valence band of monolayer InSe exhibits a sombrero-like shape (occasionally seen in inversion layers of strained p-channel Si, Ge, and III-V field effect transistors [58]). The resulting large density of states, actually singular at the band edge, results in an increased conductivity in p-FETs [59].

We have calculated the phonon dispersion and the interaction between electrons and phonon using density functional perturbation theory and interpolated into a fine mesh using maximally localized Wannier functions. We have also calculated the electron scattering rates. By separating the contribution of the short-range (deformation potentials) and long-range (Fröhlich and piezoelectric) interactions, we found that long-range interactions dominate transport. Of particular interest is the fact that piezoelectric scattering remains strong even at room temperature, in contrast to bulk III-V compound semiconductors, due to a lack of inversion symmetry in monolayer InSe.

Using the full-band Monte Carlo method, we calculated the low-field electron mobility and velocity-field characteristics of monolayer InSe. We found an electron mobility of 110 cm^2^V^−1^s^−1^ when accounting for all electron-phonon interactions and values of 188 cm^2^V^−1^s^−1^ and 365 cm^2^V^−1^s^−1^ when considering separately the contributions of long-range and short-range interactions, respectively. These values are comparable, or even marginally better, than the electron mobility calculated or measured in many other TMDs or most important, in ultra-thin Si bodies. We have also compared the dependence of exchange and correlation functionals used in the DFT calculations on the transport properties of the material. We obtained an electron mobility of 154 cm^2^V^−1^s^−1^ when using the LDA exchange-correlation functional. We do not observe much dependence on the exchange-correlation functional used to perform DFT calculations for InSe as compared to the dependence in WS_2_ [53], thanks to the reduced role played by the intervalley scattering at low fields.

Considering the relatively large electron mobility we have calculated here and also the observed stability of monolayer InSe nanodevices [31] in air without any fast degradation, contrary to what has been observed for black phosphorous [60], one may justifiably list monolayer InSe among the many possible promising materials that may one day help very-large-scale integration (VLSI) technology. Its strong piezoelectric properties and the very strange sombrero-like structure of the valence band are additional properties that deserve further study for possible application in flexible electronics, optoelectronics, and electromechanical systems.

## Figures and Tables

**Figure 1 materials-12-04210-f001:**
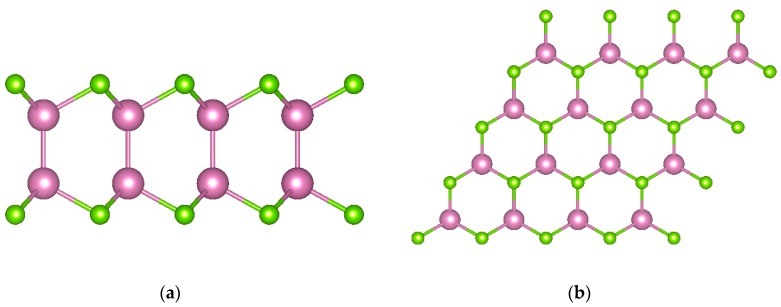
(**a**) Side view and (**b**) top view of monolayer InSe. Purple and green spheres represent In and Se atoms, respectively.

**Figure 2 materials-12-04210-f002:**
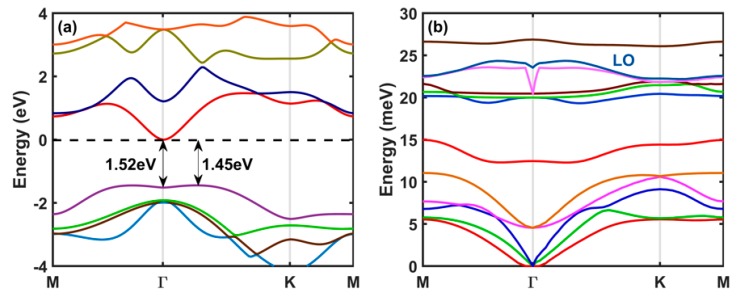
(**a**) Calculated band structure for monolayer InSe. (**b**) Calculated phonon dispersion of monolayer InSe along high-symmetry directions of the first Brillouin zone using the GGA-PBE exchange-correlation functional.

**Figure 3 materials-12-04210-f003:**
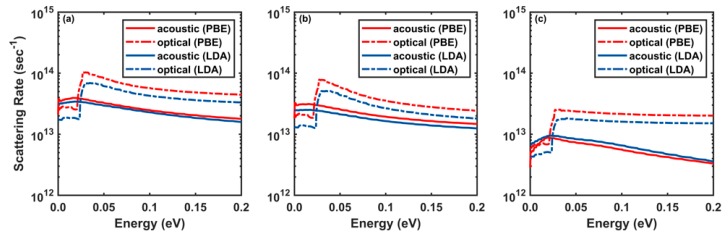
Comparing the room-temperature (T = 300 K) electron scattering rate plotted as a function of the electron energy calculated by considering (**a**) both long-range and short-range interactions, (**b**) only long-range interactions, and (**c**) only short-range interactions with DFT calculations performed using the GGA-PBE and local density approximation (LDA) exchange-correlation functionals.

**Figure 4 materials-12-04210-f004:**
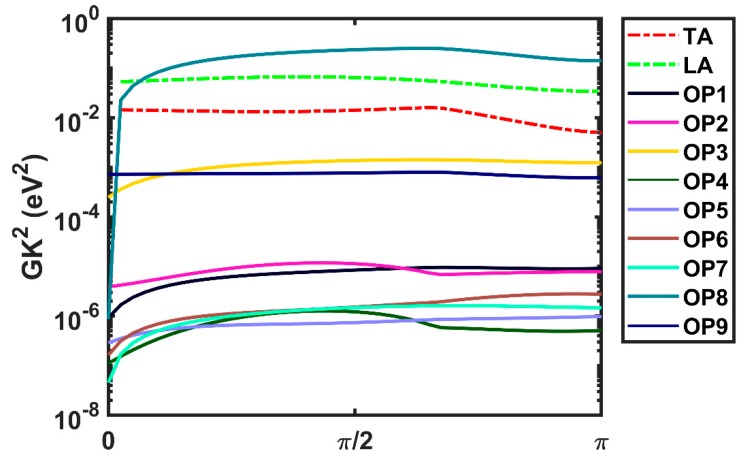
The electron-phonon matrix elements, 25 meV above the conduction band minimum, calculated as a function of the angle between the initial wavevector (oriented along the Γ-K direction) and the final wavevector for various phonon modes in monolayer InSe at 300 K using the GGA-PBE exchange-correlation functional. The matrix elements with ZA phonons vanish to first order and are not shown. Acoustic phonons are indicated by dashed lines whereas optical phonon modes are indicated by solid lines (color online).

**Figure 5 materials-12-04210-f005:**
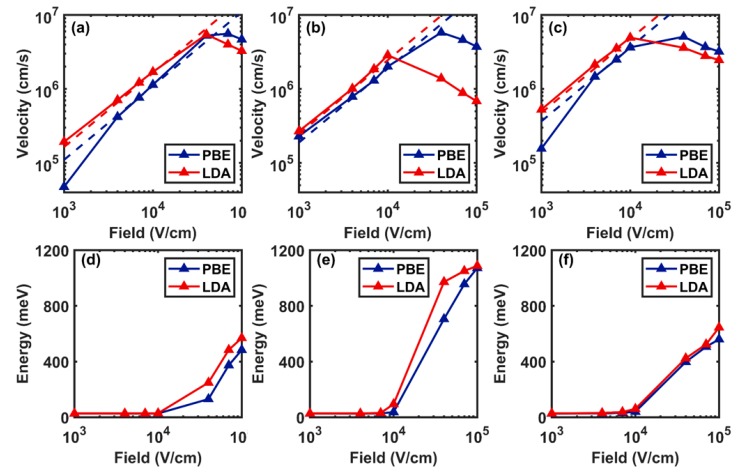
Velocity-field characteristics of electrons for the GGA-PBE and LDA cases considering (**a**) total matrix elements, (**b**) long-range interactions only, (**c**) short-range interactions only, and their corresponding (**d**–**f**) average energy-field plot calculated using Monte Carlo method at room temperature (T = 300K).

**Figure 6 materials-12-04210-f006:**
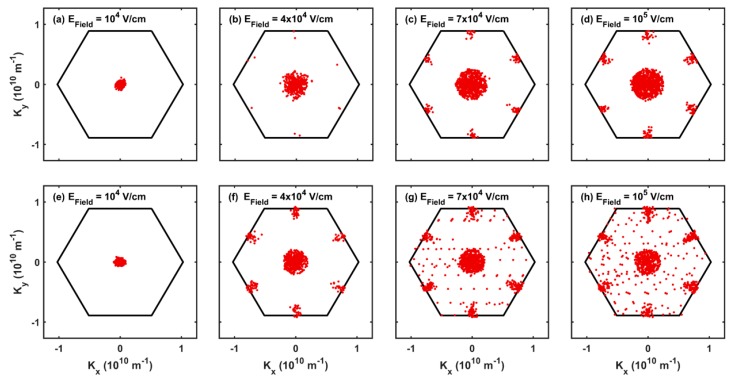
Room temperature distribution of electrons in the first Brillouin zone under the acceleration caused by a homogeneous electric field of strength ranging from (**a**–**d**) low to high for the GGA-PBE and LDA (**e**–**h**) cases. At low fields, the electrons populate only the Γ-valley. With increasing magnitude of the electric field, the electrons gain enough energy to scatter to the satellite M-valleys.

**Table 1 materials-12-04210-t001:** Structural and electronic parameters for monolayer InSe, relaxed and calculated in this work using DFT with Optimized Norm Conserving Vanderbilt (ONCV) pseudopotentials and a GGA-PBE exchange-correlation functional, compared to previously published first-principles results [29,33,34,35].

Parameter	This Work	Previous First-Principles Results
Lattice Constant (Å)	4.05	4.09 [33], 3.95 [29], 4.04 [34], 4.09 [35]
In-In bond length (Å)	2.82	2.82 [33], 2.81 [34], 2.83 [35]
In-Se bond length (Å)	2.67	2.69 [33], 2.62 [29], 2.69 [34]
Band gap (eV)	1.45	1.41 [33], 1.67 [29], 2.18 [34], 1.44 [35]

**Table 2 materials-12-04210-t002:** Computational parameters for density functional theory (DFT) and density functional perturbation theory (DFPT) calculations.

Parameters	QE
Kinetic energy (***E***_k_) cutoff	60 Ry
Charge density cutoff	240 Ry
Ionic minimization threshold	10^−6^ Ry
Self-consistent field threshold	10^−12^ Ry
***k*****-**point mesh	12 × 12 × 1

**Table 3 materials-12-04210-t003:** Phonon-limited electron mobility of monolayer InSe at room temperature (T = 300K).

Type of Scattering Processes	µ_e_ (cm^2^V^−1^s^−1^)	
	GGA-PBE	LDA
Total	110	154
Long-range interaction only	188	225
Short-range interaction only	365	465

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
