# Peer review of "Monte Carlo Study of Electronic Transport in Monolayer InSe"

_materials, 2019, doi:10.3390/ma12244210_

Round 1

Reviewer 1 Report

The problem of electron/hole mobility in InSe is considered. The paper compares the new DFT/MC results with the previous results and solves the problem of their discrepancy. I found the paper interesting, but I have a few remarks I would like the Authors to address:

Technical remarks:

1. Lines 170-171. A very fine mesh is used in BS calculations. Is it really necessary? Can the Authors show for example how the carrier velocity converges with finer mesh?

2. Lines 194-196. It is written that SOC has negligible effect on the material's properties. Can the Authors show the exact difference between calculations with and without SOC taken into account?  

3. Paragraphs 3.3-3.5. The Authors compare the results for the calculations using LDA and GGA. I do not really see the point. It is well-known that GGA overcomes many of the LDA's deficiencies and it is currently one of the standard choices (along with hybrid functionals). Certainly, it has bigger computational cost, but the modern supercomputers are fast enough to do this kind of calculations (especially for a small unit cell like the one used in the paper) in reasonable time. Is it not a step back?

A substantive issue:

4. The problem considered here - the charge mobility in 2D materials - is now very common. A lot of papers considering different materials are published. Unfortunately, most of the papers - and this one as well - do not consider the impact of the surface. If we really think about any application of the 2D materials we have to take this problem into account. Interactions with a surface will affect both geometrical and energy properties of InSe and thus it will definitely alter the carrier mobility. And for a monolayer this effect will be of significant importance as there will be no 'buffer layer' between InSe and the surface. Can the Authors justify the 'InSe in vacuum' model?  

Reviewer 2 Report

Page 4, Lines 113-118

The band structure of monolayer InSe is obtained from density functional theory, as implemented in the Quantum ESPRESSO (QE) [38][39] software package, with both the Perdew-Burke-Enzerhoff generalized-gradient approximation (GGA-PBE) [40] and local density approximation (LDA) [41] for the exchange-correlation functional, and the Optimized Norm-Conserving Vanderbilt (ONCV) pseudopotentials [42].

Page 6, Lines 191-194

In Fig. 2a, we show the calculated band structure of monolayer InSe along the high symmetry directions in the first Brillouin zone. Monolayer InSe is an indirect band gap material with a band gap of 1.45 eV with the conduction-band minimum at Γ and valence-band maxima along the Γ-K direction.

Is the band structure plotted in Fig.2a calculated using PBE or LDA?

Page 6, Line 210

The FBZ abbreviation is not explained before.

Question about band gap value (1.45 eV).

It is well-know that band gap value (PBE and LDA approximations) is slightly underestimate the experimental value that is common situation for DFT since the theory suffers from the well-known band gap underestimation problem.

I think that it is necessary to calculated band structure using Nonlocal exchange-correlation functional, PBE0 or HSE06, and compare obtained results with published before, for example in

https://www.nature.com/articles/s41535-018-0089-0

See Table 1 or Fig. 2a

Reviewer 3 Report

The manuscript under the title: "Monte Carlo study of electronic transport in monolayer InSe" is only partly coherent with Materials journal.

The authors present original research works. The aim of article is clearly formulated. The research methodology is well described. The authors present interesting and up-to-date topic. The organization of the article is appropriate. The abstract is sufficiently informative. Overall, the paper is well prepared. The some editorial improvements is required, including: reference style in text and removing subscription under the Figure 5.

Additionally, please also explain the source of "this work" in Table 1 (more detailed comment is required under this table).

Round 2

Reviewer 1 Report

I am satisfied with the Authors' responses and the reviewed manuscript. I recommend to publish the paper as is.

Reviewer 2 Report

The authors have satisfactorily responded to all my questions and made the necessary changes to the manuscript. I think that this article can be accepted in the present form.